# Evolution of the Diagnosis and Treatment of Primary Hyperparathyroidism

**DOI:** 10.3390/jcm12052057

**Published:** 2023-03-06

**Authors:** Enrico Battistella, Luca Pomba, Riccardo Toniato, Marta Burei, Michele Gregianin, Sara Watutantrige Fernando, Antonio Toniato

**Affiliations:** 1Endocrine Surgery Unit, Department of Surgery, Veneto Institute of Oncology, IOV-IRCCS, Via Gattamelata 64, 35128 Padua, Italy; 2School of Medicine, University of Padua, Via Giustiniani 2, 35128 Padua, Italy; 3Department of Nuclear Medicine, Veneto Institute of Oncology, IOV-IRCCS, Via Gattamelata 64, 35128 Padua, Italy; 4Familial Cancer Unit, Veneto Institute of Oncology, IOV-IRCCS, Via Gattamelata 64, 35128 Padua, Italy

**Keywords:** parathyroid surgery, primary hyperparathyroidism, intraoperative PTH-assay, mini-invasive parathyroidectomy, indocyanine green angiography, indocyanine green fluorescence

## Abstract

This study aims to present the evolution of our center’s approach to treating primary hyperparathyroidism (PHPT) from diagnosis to intraoperative interventions. We have also evaluated the intraoperative localization benefits of indocyanine green fluorescence angiography. This retrospective single-center study involved 296 patients who underwent parathyroidectomy for PHPT between January 2010 and December 2022. The preoperative diagnostic procedure included neck ultrasonography in all patients, [99mTc]Tc-MIBI scintigraphy in 278 patients, and, in 20 doubtful cases, [18F] fluorocholine positron emission tomography (PET) computed tomography (CT) was performed. Intraoperative PTH was measured in all cases. Indocyanine green has been administered intravenously since 2020 to guide surgical navigation using a fluorescence imaging system. The development of high precision diagnostic tools that can localize an abnormal parathyroid gland in combination with intra-operative PTH assay (ioPTH) enables the surgical treatment of PHPT patients with focused approaches and excellent results that are stackable with bilateral neck exploration (98% of surgical success). Indocyanine green angiography has the potential to assist surgeons in identifying parathyroid glands rapidly and with minimal risk, especially when pre-operative localization has failed. When everything else fails, it is only an experienced surgeon who can resolve the situation.

## 1. Introduction

Parathyroid glands (PG) are small, oval-shaped endocrine glands typically found behind the thyroid lobes. They regulate calcium through the production of parathyroid hormone (PTH). Primary hyperparathyroidism (PHPT) is a disease of the parathyroid gland (PG), representing the third endocrinopathy following diabetes mellitus and thyroid disease. The incidence rate is approximately 25 per 100,000 people and, as in other endocrine diseases, is more common in females than in males, with a ratio of 3:1 and an incidence peak in the early post-menopausal phase [1,2]. PHPT may be caused by parathyroid adenomas (in >80% of cases), multiple adenomas, parathyroid hyperplasia (about 15%), or parathyroid carcinoma (<1%) [3].

As the only curative treatment for PHPT, parathyroidectomy is recommended for symptomatic patients as well as those who are asymptomatic and at risk of disease progression or have subclinical evidence of end-organ effects [4]. Classic preoperative localization techniques include ultrasonography and nuclear scintigraphy, while the intraoperative procedures we have employed are intraoperative PTH assay (IO PTH-assay) and indocyanine green (ICG) angiography [5]. Successful preoperative and intraoperative localization of the parathyroid gland enables a minimally invasive approach and eliminates the need for routine 4-gland exploration, thereby decreasing the risk of bilateral recurrent laryngeal nerve injury or hypoparathyroidism.

In this case series, we describe our approach to the disease, from diagnosis through surgical treatment, including the intraoperative procedures performed. 

## 2. Materials and Methods

We used a computerized endocrine surgery registry to record the demographic and clinical data of 296 patients who had surgery for PHPT at our institution. In accordance with the guidelines of the American Association of Endocrine Surgeons, those who were eligible for surgery and included in our study were symptomatic and asymptomatic PHPT patients aged < 50 years with persistently elevated serum calcium, vertebral fracture, nephrolithiasis or nephrocalcinosis, and creatinine clearance of <60 mL/min [6]. Patients with secondary and tertiary hyperparathyroidism were excluded.

Preoperative investigations, such as ultrasonography (US) and [99mTc]Tc-MIBI scintigraphy, were performed in order to successfully localize the pathological PG. In cases with no or doubtful localization, [18F] fluorocholine positron emission tomography (PET) computed tomography (CT), which was introduced at our institution in 2017, was used as another important diagnostic tool to localize PG. 

All operations were performed by the same endocrine surgery team. Minimally invasive parathyroidectomy (MIP) was the standard surgical procedure, with a focused dissection if preoperative imaging modalities indicated laterality; otherwise, bilateral neck exploration (BE) was performed. Mini-incision parathyroidectomy and mini-invasive video-assisted parathyroidectomy were conducted as part of the MIP procedure.

PTH assay was performed on all patients on the day of hospitalization and 10 min after abnormal PG removal to provide the necessary assurance that a focused parathyroidectomy had been effectively undertaken.

Since the year 2020, intraoperative fluorescence imaging with indocyanine green (ICG) was used for the purposes of this study. After the suspected lesion was identified and exposed, 3 mL of ICG (2.5 mg/mL of ICG dissolved in sterile water) was administered as an intravenous bolus, and the fluorescence-capable laparoscope was directed into the operating field. Typically, exposed parathyroid adenomas exhibited fluorescent enhancement one minute after injection. The PG was excised once fluorescence with ICG was confirmed. 

Surgical failure was defined as the persistence of hypercalcemia and of the PTH value above normal range. 

## 3. Results

Our study involved 296 patients diagnosed with PHPT (71 males, 225 females, M:F = 1:3), each of whom underwent surgery between January 2010 and December 2022. The average age at the time of surgery was 51.8 years (range 11–84 years). 

All patients had neck ultrasonography (US), while 278 patients (94%) also had a [99mTc]Tc-MIBI scintigraphy. Over the past 5 years, 20 patients had [18F] fluorocholine PET/CT for uncertain localization or no localized PG. Pathological parathyroids were correctly identified in 228 (77%) of the 296 patients by neck US, in 189 (68%) of the 278 patients by scintigraphy and in 19 (99%) of the 20 patients by [18F] fluorocholine PET/CT. 

The surgical approach was chosen from bilateral neck exploration (BE) and minimally invasive parathyroidectomy (MIP) based on the results of preoperative investigations. 

MIP was performed in 210 cases (71%), with mini-incision parathyroidectomy at the outset in 160 cases (76%) and mini-invasive video-assisted parathyroidectomy in 50 cases (24%). In four patients (2%), conversion to bilateral exploration was necessary due to the inability to identify the abnormal parathyroid gland despite preoperative imaging. BE was chosen in 86 cases (29%) as the gold standard technique in the absence or doubtful localization, multiple adenomas, parathyroid hyperplasia, concomitant thyroid disease, and clinical suspicion of parathyroid carcinoma. 

PTH assay was performed on all patients on the day of hospitalization and 10 min after abnormal PG removal. The Io-PTH assay showed a reduction of more than 70% in 242 cases (82%), and a decrease of 50–70% in 44 cases (15%). There was no significant decrease in Io-PTH in 10 cases (3%). 

In all patients with a reduction in PTH value of more than 50% (286 patients, 87%), we noted that calcium levels were normalized on the first postoperative day, with normal PTH levels and normal calcium serum ranges. In five patients who had no intraoperative decrease, a decrease in normalized calcium levels was observed in the second post-operative day.

ICG has been administered to a total of 18 patients undergoing parathyroidectomy since the year 2020. ICG has been used to compare preoperative localization, visual identification, and the correspondence of pathological PG with ICG angiography in 12 patients (100%). A parathyroid adenoma was sought to be identified in six patients by means of ICG fluorescence imaging during a BE, following an ultrasonography and a preoperative [99mTc]Tc-MIBI scintigraphy scan that failed to localize the disease or resulted in uncertain localization: in five patients, the procedure was successful, whereas in one patient, the parathyroid adenoma was not identified. 

The reported complications included one case each of neck hematoma and transient recurrent laryngeal nerve injury (0.3%). There was no evidence of a transient state of hypocalcemia in the first postoperative day.

The average length of stay (LOS) was 2 days (1–3 days).

Surgical failure (defined as the persistence of hypercalcemia and hyperparathyroidism) occurred in five cases (1.7%). All unsuccessful procedures arose when preoperative localization was uncertain.

Histological reports revealed 248 single adenomas (84%), 30 multiple adenomas (10%), 12 hyperplasia (4%), 3 atypical adenomas (1%), and 3 parathyroid carcinomas (1%). Parathyroidectomies, homolateral thyroid lobectomies, and lymph node dissections were performed on patients with parathyroid carcinoma.

## 4. Discussion

PHPT is an endocrine disorder characterized by an inappropriate excessive production of PTH that results in hypercalcemia. A rare type of PHPT is normocalcemic hyperparathyroidism, presenting normal serum calcium levels but an elevated PTH concentration. It is typically limited to a single adenoma, but multiglandular disease occurs in 10% to 15% of patients, as multiple adenomas or hyperplasia, while parathyroid carcinoma accounts for less than 1% of cases. There are two types of PHPT: the sporadic form (with no familial history) and the hereditary form, with an incidence of no more than 10% in the PHPT population (MEN I, MEN IIa, MEN IV, familial hypocalciuric hypercalcemia, and neonatal severe PHPT) [2,3]. At diagnosis, 85% of patients were completely asymptomatic or mildly symptomatic, the so-called biochemical hyperparathyroidism [1,2]. Parathyroidectomy is the definitive treatment for PHPT, but this procedure requires an adept surgeon due to the small dimensions and variable localization of PG. Although parathyroid operation by an experienced surgeon results in an over 90–95% success rate, the success rate of a surgeon doing fewer than 10 parathyroidectomies per year is about 70% [7]. Persistent or recurrent PHPT can be caused by incomplete resection of hyperplastic parathyroid glands or failure to recognize and localize the parathyroid adenoma [8]. The typical site of the superior parathyroid glands is within a 1–2 cm radius from the point of intersection of inferior thyroid artery with the recurrent laryngeal nerve, within the interstitial space between the postero-medial aspect of the upper thyroid pole and the trachea while the inferior glands lie within 1.5 cm of the lower pole of the thyroid [9]. Despite this, the incidence of ectopic parathyroid glands is approximately 16% [10]. Ectopic parathyroid glands can be retroesophageal, between the trachea and the esophagus, behind the pharynx, and inside the thyroid gland, the thymus, and the mediastinum tissue, accounting for 10–20% of all parathyroid glands [11,12]. 

Preoperative imaging is indispensable for the surgeon to determine the optimum surgical strategy. Diagnostic imaging should be focalized in order to correctly locate the pathological parathyroid. Neck ultrasound (US) is the most common technique used for localization, because it is non-invasive and the least expensive technique. Normal glands are infrequently visible on ultrasound, while enlarged parathyroids appear as homogeneous, hypoechoic with a mark shape, or are thin “like a cigar” (8–15 mm) with well-defined boundaries [13]. The literature reports a US sensitivity of 85%, with a range of 60–95% depending on the gland’s dimension and the radiologist’s experience [5,14]. The US sensitivity dropped to 35% for patients with multiglandular disease and to 16% for those with double adenomas. Another problem of US is the limited in its ability to assess ectopic parathyroid glands [15].

[99mTc]Tc-MIBI scintigraphy is another important diagnostic tool for localizing parathyroid glands. Sestamibi is a lipophilic cation that accumulates in the mitochondria and, owing to the abundance of mitochondria in the oxyphil cells, there is high accumulation of [99mTc]Tc-MIBI in hyperfunctioning parathyroid glands [16]. This procedure has a sensitivity of between 69 and 91% for detecting abnormal parathyroid glands. However, it can only image pathological PG because normal PG are too small to be detected [14]. The advantage of the radionuclide parathyroid imaging over the US lies in the identification of the ectopic glands, it also enables an easier recognition of typically located parathyroid glands in the backgrounds of thyroid glands. The disadvantage of the [99mTc]Tc-MIBI scintigraphy is the potential of false positives; for example, follicular and Hurtle cell thyroid neoplasm are prone to sestamibi accumulation, and the same holds true of thyroiditis and lymphadenopathy [17].

Boudousq et al. suggest using [18F] fluorocholine PET/CT as a first-line examination in preference to US and [99mTc]Tc-MIBI scintigraphy. They have reported that 18F-FCH PET with computed tomography (CT) is more accurate in detecting parathyroid glands than conventional morphological and functional imaging systems (US + scintigraphy); with a superior sensitivity of 96%, it generates a lower radiation dose than MIBI scintigraphy and, ultimately, it is the best examination for identifying ectopic adenomas [18,19]. Other authors maintain that [18F] fluorocholine PET/CT should be considered only in cases where the location of abnormal PG is uncertain as a second-line examination, when there is disagreement or negative results with US and [99mTc]Tc-MIBI scintigraphy given its cost-effectiveness and inaccessibility [20,21]. We have performed twenty [18F] fluorocholine PET/CT at our institution since 2017 as a second-line examination (Figure 1).

Another diagnostic instrument is 4D-CT, a new modality for parathyroid imaging that is based on the features of perfusion of parathyroid adenomas. The fast wash-in and wash-out of adenomas can be analyzed using a rapid sequence of images, resulting in highly-detailed multiplanar images that can reveal abnormal parathyroid tissue, differentiating it from thyroid tissue and lymph-nodes. Studies in the literature support recourse to this method in the event of neck reoperation. Multiglandular disease or ectopic parathyroid glands can be detected with 4D-CT, which has an 86% right primary positioning of abnormal parathyroid glands. The disadvantage is exposure to a high amount of radiation, which is why it should only be used in difficult cases [22,23].

It is important to include abdominal imaging (abdominal X-ray or ultrasound) and vertebral imaging to complete the evaluation of the disease. Indeed, many studies have revealed that patients with asymptomatic PHPT have vertebral involvement as well as renal stones or nephrocalcinosis [24].

The most prevalent approach to localize abnormal parathyroid gland in a patient with a de novo diagnosis of PHPT follows the approach suggested by Kunstman et al. [25]. In our department, the gold standard for preoperative localization is US combined with [99mTc]Tc-MIBI scintigraphy to minimize negative or equivocal identification of each single diagnostic tool (only US or only scintigraphy) [18]. The common starting point is US, because it is widely available and less expensive. Scintigraphy is less operator-dependent than ultrasound and is an effective way to visualize ectopic gland, while US allows simultaneous evaluation of the thyroid gland. If the result is definitive, we usually propose to the patient a mini-invasive parathyroidectomy. If both tests are contradictory or inconclusive, we suggest a [18F] fluorocholine PET/CT [25]. Cuderman et al. demonstrated in their study a sensitivity of 92% for [18F] fluorocholine PET/CT compared to a 65% for conventional imaging [26]. They have also highlighted the importance of detecting lesions <10 mm in diameter. As reported above, the potential disadvantages include higher costs, the uptake by inflammatory lymph nodes, and thyroid nodules as possible false positives [26]. We do not perform 4D-CT because is not available in our institute. 

Pre-operative localization of the pathological parathyroid gland is essential to permit a minimally invasive procedure; otherwise bilateral neck exploration is necessary. A minimally invasive approach has been used on 210 patients (71%) since 2010, with a surgical success rate of 98% (206 patients); only in 4 patients was conversion to BE (2%) necessary to find the parathyroid adenoma. Our institution has systematically implemented this procedure for PHPT surgery. Successful preoperative localization is the principal eligibility criterion for a focused approach, as the abnormal parathyroid gland is approached with a focused mini-incision [27,28,29].

In each operation, the intraoperative PTH-assay (Io-PTH assay) was employed to confirm the excision of the pathological parathyroid tissue. The literature reports that the Io-PTH assay has a clinical sensitivity of 85% when a reduction of more than 50% of the PTH starting value is observed in a blood sample five minutes after surgical excision, while sensitivity reaches 97% after 10 min. For this reason, we preferred the 10-min assay after surgical excision [30,31]. Due to the molecule’s short half-life, the variation in PTH value can be considered significant in just a few minutes after parathyroidectomy, being able to substantially reduce the time required for traditional exploration of the neck region. Some authors have reported that the PTH value decreases after only five minutes from the removal of the parathyroid gland [32].

The inability to localize the PG with preoperative imaging necessitates an inspection of all the PG, which includes the area extending from the carotid bifurcation to the mediastinum. The exploratory technique includes the section of the upper thyroid peduncles, the medialization of thyroid lobes, and the section of the medium thyroid vein. In addition, a trained eye is required to recognize a normal PG, which appears tan in color with a diameter of 4–6 mm and is surrounded by an adipose capsule. 

In this setting, indocyanine green (ICG) fluorescence angiography could be useful, and we began using it in 2020 (Figure 2). ICG is an inert, non-toxic, organic dye. Following intravenous injection, it circulates through the intravascular space bound to plasma proteins until it is cleared primarily through the hepatobiliary system. ICG is used in many areas of medicine as a marker for identifying anatomical structures and evaluating tissue, such as cholangiography, perfusion assessment of gastrointestinal anastomoses, adrenalectomy, or real-time lymph node mapping [33]. 

ICG is a nonselective agent, which constitutes a limitation of its intraoperative application in PGs detection since it does not target parathyroid parenchyma specifically. However, because PGs receive a higher amount of blood compared to surrounding tissues, they emit a much stronger fluorescent signal, which consequently presents the exact localization of the PGs. De Long et al. have shown that ICG fluorescence angiography has the potential to assist surgeons in identifying parathyroid adenomas rapidly with minimal risk. In their study, out of the 54 patients who had a preoperative sestamibi scan, a parathyroid adenoma was identified in 36, while 18 failed to localize. Of the 18 patients who failed to localize, all 18 patients (100%) had an adenoma that fluoresced on indocyanine green imaging [34].

At the beginning of our experience, we used ICG to verify correspondence between preoperative localization and the naked eye, and ICG angiography to identify the pathological PG. This was then carried out in six cases with no or doubtful preoperative localization. In these cases, pathological tissue was not detected by visual inspection and palpation, but rather by ICG angiography in five cases. Once the thyroid lobe is retracted medially, the first injection dose is administered intravenously, and uptake in the PG is typically seen after 20–60 s. In accordance with the literature, we have confirmed that this technique can be a useful adjunct for patients with non-localizing preoperative imaging results [35,36]. Zaidi et al. have demonstrated the feasibility of localizing parathyroid glands intraoperatively with ICG fluorescence angiography. They concluded that ICG can localize parathyroid glands reliably during parathyroidectomy and, in addition, allow for an assessment of parathyroid perfusion in patients undergoing subtotal resection [37].

The real advantage we found was the possibility to perform a focused dissection without the risk of bilateral injury to the recurrent laryngeal nerve and postoperative neck hematoma. 

Furthermore, a pre-operative localization and an intra-operative localization are primary tools to avoid persistent or recurrence PHPT related to inadequate resection of hyperplastic parathyroid glands, failure to recognize and localize parathyroid adenomas, presence of more than one adenoma, error in diagnosis, and surgical inexperience. Re-exploration after a previous parathyroid operation is, unfortunately, associated with a higher morbidity and a lower success rate [8,38]. 

## 5. Conclusions

Parathyroidectomy offers a unique possibility to address PHPT and avoid fatal complications [39]. The improvement in diagnostic tools makes the minimally-invasive approach the technique of choice when localization is well-defined. In our Department, US combined to [99mTc]Tc-MIBI scintigraphy continue to be the gold standard for localization of parathyroid adenomas. In case of doubtful or negative localization and in case of suspected hyperplasia or multiglandular adenomas, additional imaging as [18F] fluorocholine PET/CT. However, bilateral neck exploration remains the gold standard technique when preoperative imaging is inconclusive. A helpful solution in this tricky situation is ICG fluorescence angiography, which should be considered as an adjunctive localization method during parathyroid surgery. Furthermore, the intraoperative PTH-assay (Io-PTH assay) should be employed to confirm the excision of the pathological parathyroid tissue. Pre-operative and intraoperative localization have to be extremely detailed to avoid recurrent or persistent PHPT and a surgical re-exploration. Diagnostic imaging should be focalized in order to identify the pathological parathyroid and to reduce the rate of unsuccessful surgery. PHPT surgery also represents a clinical challenge to expert surgeons and thanks to our diagnostic approach (pre-operative and intra-operative), our success rate (1.7%) is lower than the reported one in Literature (5–10%) [8]. 

## Figures and Tables

**Figure 1 jcm-12-02057-f001:**
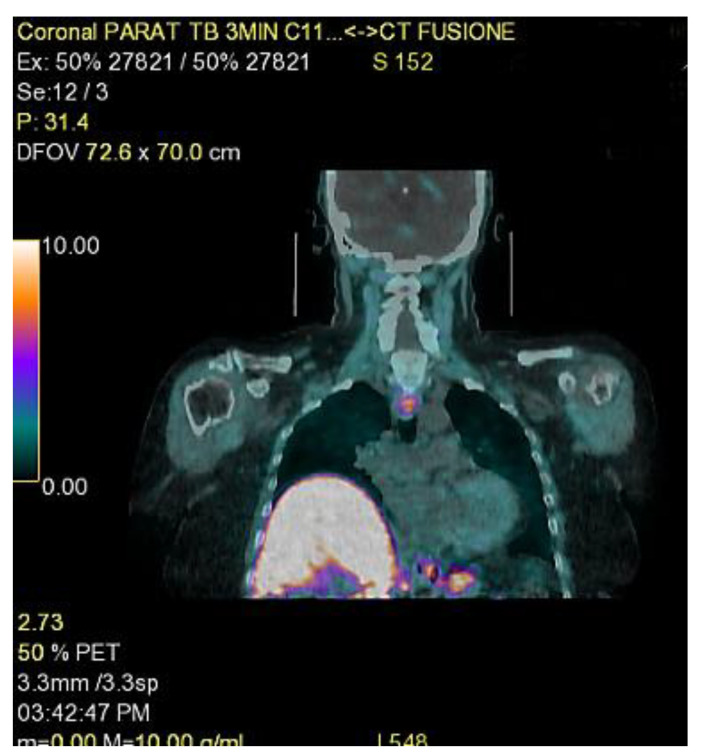
Mediastinic parathyroid adenoma localized with [18F] fluorocholine PET/CT (US and scintigraphy were not useful).

**Figure 2 jcm-12-02057-f002:**
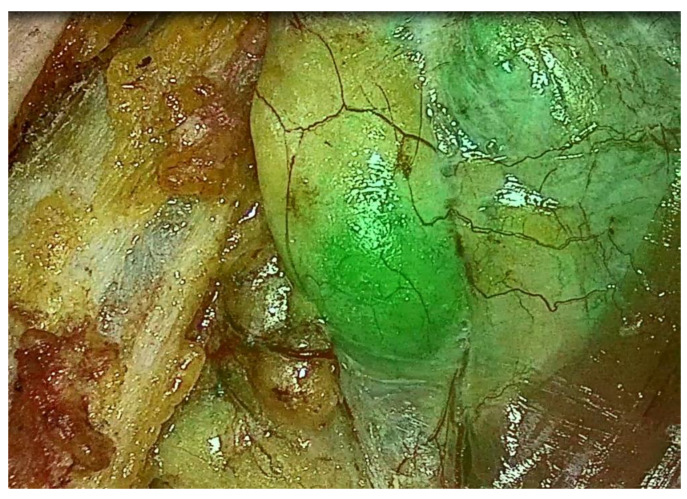
ICG angiography: parathyroid adenoma identified intra-operatively.

## Data Availability

Data were retrieved from patient medical records.

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
