# Peer review of "Evolution of the Diagnosis and Treatment of Primary Hyperparathyroidism"

_jcm, 2023, doi:10.3390/jcm12052057_

Round 1
Reviewer 1 Report
The author studied the value ultrasound, MIBI, PET, and intraoperative fluorescence imaging with ICG for locatization of abnormal parathyroids.The topic is interesting and significant, and the results are credible. However, the comparison of these localization technique need to be shown in more details. I think not all these 4 techniques are needed for all patients, therefore, how to chose these technique? For example, what is the sensitivity and accuracy for these 4 localization techniques? If ultrasound and MIBI are consistent, any other technique is needed? If yes, in what circumstance? If not, why? In what circumstance the fluorescence imaging with ICG is needed? these need to be shown and explained in the results and disccusion.
Author Response
ANSWER TO REVIEWER 1
- The author studied the value ultrasound, MIBI, PET, and intraoperative fluorescence imaging with ICG for locatization of abnormal parathyroids.The topic is interesting and significant, and the results are credible. However, the comparison of these localization technique need to be shown in more details. I think not all these 4 techniques are needed for all patients, therefore, how to chose these technique? For example, what is the sensitivity and accuracy for these 4 localization techniques? If ultrasound and MIBI are consistent, any other technique is needed? If yes, in what circumstance? If not, why? In what circumstance the fluorescence imaging with ICG is needed? these need to be shown and explained in the results and discussion.
- We have extended the discussion from line 211 to 226 to explain how to chose the most appropriate imaging tool.
“The most prevalent approach to localize abnormal parathyroid gland in a patient with a de novo diagnosis of PHPT follows the suggested approach by Kunstman et al. (25) In our department, the gold standard for preoperative localization is US combined with [99mTc]Tc-MIBI scintigraphy to minimize negative or equivocal identification of each single diagnostic tool (only US or only scintigraphy). (18) The common starting point is US because it is widely avaiable, less expensive. Scintigraphy is less operator-dependent than ultrasound and is an effective way to visualize ectopic gland while US allows simultaneous evaluation of the thyroid gland. If the result is definitive, we usually propose to the patient a mini-invasive parathyroidectomy. If both tests are contradictory or inconclusive, we suggest a [18F] fluorocholine PET/CT. (25) Cuderman et al. demonstrated in his study a sensitivity of 92% for [18F] fluorocholine PET/CT compared to a 65% for conventional imaging. (26) They have also highlighted the importance to detect lesions < 10 mm in diameter. As reported above, the potential disadvantage are higher costs, the uptake by inflammatory lymph nodes and thyroid nodules as possible false positive. (26) We do not perform 4D-CT because is not available in our Institute. “
The sensitivity for these 4 localization techniques have been reported in the original manuscript at lines 166-168, line 175, line 187 and line 204.
The indication to use ICG has been shown at line 276-278.
“In accordance with the literature, we have confirmed that this technique can be a useful adjunct for patients with non-localizing preoperative imaging results. “
We performed an english revision as required
Reviewer 2 Report
The authors presented their experiences with diagnosis and treatment of hyperthyroidism which in general not complicated, but is some cases may constitute a clinical challenge.
Main comment:
The kind of novelty in this manuscript constitute the use of intraoperative fluorescence imaging with indocyanine green (ICG) for parathyroid adenoma (PA) visualization which is described in literature for relatively short period of time. However in the part concerning ICG use the authors actually show a very small number of patients (6) in whom the procedure with ICG was clinically justified (performed in situation of no or doubtful adenoma visualization), although here the percentage of positive results was high. Rest of the described methods of PA visualization are very well known and add nothing new to current knowledge.
Minor comments:
- In the abstract, the ioPTH abbreviation has no explanation
- Please correct the names of radiopharmaceuticals according to Enam nomenclature guidelines (https://www.eanm.org/publications/guidelines/nomenclature/)
- Please remove results from discussion (lines 190-193) and complete lacking results from those lines in results section
- There is no information if current study obtained consent of the bioethics committee
Author Response
ANSWER TO REVIEWER 2
- Main comment:
The kind of novelty in this manuscript constitute the use of intraoperative fluorescence imaging with indocyanine green (ICG) for parathyroid adenoma (PA) visualization which is described in literature for relatively short period of time. However, in the part concerning ICG use the authors actually show a very small number of patients (6) in whom the procedure with ICG was clinically justified (performed in situation of no or doubtful adenoma visualization), although here the percentage of positive results was high. Rest of the described methods of PA visualization are very well known and add nothing new to current knowledge.
- We consider PHPT surgery a real clinical challenge and for this reason we proposed in the manuscript the approach used in our department from diagnosis to the surgical strategies. This approach permits an unsuccessful rate lower than the reported one in Literature and it is described in the discussion section.
Minor comments:
-In the abstract, the ioPTH abbreviation has no explanation
- Now ioPTH is explained.
-Please correct the names of radiopharmaceuticals according to Enam nomenclature guidelines (https://www.eanm.org/publications/guidelines/nomenclature/)
- All the names of radiopharmaceuticals were corrected in [18F] fluorocholine PET/CT and [99mTc]Tc-MIBI scintigraphy
-Please remove results from discussion (lines 190-193) and complete lacking results from those lines in results section
- We have solved this mistake.
-There is no information if current study obtained consent of the bioethics committee
- Ethical approval was not required for this retrospective study in our country but the study was conducted respecting the Helsinki Declaration. The patients have given informed consent to use anagraphic and clinical data for the research.
Round 2
Reviewer 1 Report
My questions are fully answered.